# Cluster Analysis of Medicinal Plants and Targets Based on Multipartite Network

**DOI:** 10.3390/biom11040546

**Published:** 2021-04-08

**Authors:** Namgil Lee, Hojin Yoo, Heejung Yang

**Affiliations:** 1Department of Information Statistics, Kangwon National University, Chuncheon 24341, Korea; namgil.lee@bionsight.com; 2Bionsight, Incorporated, Chuncheon 24341, Korea; hojinyoo@bionsight.com; 3Laboratory of Natural Products Chemistry, Department of Pharmacy, Kangwon National University, Chuncheon 24341, Korea

**Keywords:** medicinal plants, multi-chemicals, multi-targets, multipartite network, network analysis

## Abstract

Network-based methods for the analysis of drug-target interactions have gained attention and rely on the paradigm that a single drug can act on multiple targets rather than a single target. In this study, we have presented a novel approach to analyze the interactions between the chemicals in the medicinal plants and multiple targets based on the complex multipartite network of the medicinal plants, multi-chemicals, and multiple targets. The multipartite network was constructed via the conjunction of two relationships: chemicals in plants and the biological actions of those chemicals on the targets. In doing so, we introduced an index of the efficacy of chemicals in a plant on a protein target of interest, called target potency score (TPS). We showed that the analysis can identify specific chemical profiles from each group of plants, which can then be employed for discovering new alternative therapeutic agents. Furthermore, specific clusters of plants and chemicals acting on specific targets were retrieved using TPS that suggested potential drug candidates with high probability of clinical success. We expect that this approach may open a way to predict the biological functions of multi-chemicals and multi-plants on the targets of interest and enable repositioning of the plants and chemicals.

## 1. Introduction

Plants have been used as therapeutic agents for a very long time in the human history [1,2]. Single bioactive compounds in plants have been developed as well-known commercial drugs, such as paclitaxel, vinblastine, morphine, and artemisinin [3,4]. Occasionally, single components from natural products have provided chemical scaffolds for developing more potent semi-synthetic drugs, such as rosuvastatin, tramadol, and eribulin [5]. Similar to these single compound drugs that act on a specific target, a mixture of components from the whole plants or their sub-fractions have been widely used as botanical drugs, such as Veregen^®^ and Fulyzaq^®^ [6]. However, the synergistic effects of multi-chemicals in plants on multiple targets and the underlying complex biological pathways involved are less understood.

For the purpose of developing drugs composed of multi-chemicals, the identification of the interactions between the multi-chemicals and multi-targets is important to understand the therapeutic effects as well as the side effects, i.e., the on-target and off-target effects [7,8,9,10,11,12,13,14]. Identifying the synergistic drug combinations is also critical to develop effective therapies against cancers and to improve the therapeutic efficacy [15,16,17,18]. However, the complexity of screening all possible candidate combinations of multi-chemicals and multi-targets prevents the use of the classical way of identifying the interactions via biological experiments, using in vitro and in vivo assays. Since a medicinal plant produces a large number of chemical compounds, the target identification of every single chemical is impossible for the development of botanical drugs, especially to acquire approval in the global market requiring well-controlled clinical studies [19].

In recent decades, computational approaches for the systematic analysis of drug-target interactions have become available, improving efficiency and reducing costs of studying the interactions [20,21,22,23]. Among the computational approaches, network-based approaches provide alternative methods to deal with the complex molecular interactions [24,25]. Compared with other computational methods, network-based approaches do not require three-dimensional structures of the targets or negative control samples [26]. These approaches have also been proposed for botanical drug design in search of optimal combinations of medicinal plant components and prediction of multiple target activities [27]. However, studies based on the integrated networks of medicinal plants, chemicals, and multi-targets are still rare. Most studies have focused only on drug-target networks [28] or the molecular networks of traditional Chinese medicinal plants [29].

In this study, we introduced an approach based on the integrated network of medicinal plants, chemical components, and multi-targets to analyze the complex biological roles of medicinal plants in human body. First, we constructed an integrated multipartite network of the biomedical entities including Korean medicinal plants, chemicals, and protein targets from heterogeneous databases. The databases have been curated for biomedical entities, such as genes, proteins, chemicals, and species and are publicly accessible. In particular, we employed the ChEMBL database, which is one of the largest public databases of the activities of small molecules on biological targets [30,31,32,33]. It also provides the built-in, ligand-based target prediction methods that we used to generate the links between chemicals with targets in the interaction network. We also used the database on compound ingredients extracted from Korean medicinal plants, which was released in 2007 by the Korean Traditional Knowledge Portal (http://www.koreantk.com, accessed on 1 May 2019) [34].

We then preprocessed the multipartite network and adapted the spectral co-clustering method to identify the subnetworks of closely related biomedical entities within their own structural properties, such as the average bipartite clustering coefficient [35]. The clustering results could be used for the single or combination use of Korean medicinal plants in common target domains. The hierarchical clustering analysis was followed for further identification of the plant clusters based on the similarities in their chemical composition. In the analysis, we introduced a network-based mathematical index of interactions between the medicinal plants and targets, called the target potency score (TPS). The suggested index is used for ranking the plant clusters or protein targets, and for retrieving the best subnetworks of the medicinal plants, multi-chemicals, and multi-targets customized to specific medicinal purposes. Finally, we have demonstrated the utility of the simplified multipartite network with the hierarchical clusters.

## 2. Materials and Methods

### 2.1. Raw Data Sets

We employed several heterogeneous data sets to construct a multipartite network of medicinal plants, chemicals, and targets. The data set of the medicinal plants was obtained from the Korean Traditional Knowledge Portal (KTKP, http://www.koreantk.com, accessed on 1 May 2019). The database was built by the Korean Intellectual Property Office, Ministry of Trade, Industry and Energy, Republic of Korea [34]. The KTKP data sets contain a list of 5500 Korean medicinal plants and 10,056 chemicals, and their relations. The data set of chemicals and targets was obtained from the ChEMBL database (ChEMBL25, http://ftp.ebi.ac.uk/pub/databases/chembl/target_predictions, accessed on 15 March 2019). The ChEMBL25 database contains 1,879,206 compounds, 12,482 targets, and 15,504,603 bioactivity entries from 72,271 publications. All the chemicals in the KTKP data sets were identified in the ChEMBL25 database, and the links between the chemicals and single-protein targets were generated using the prediction model reported previously [33,36], and is available on the ChEMBL25 database. Specifically, the probability of the activation of a compound against a target was estimated by the 10-uM activity cut-off model in the ChEMBL25 database, and we determined the interaction between a pair of a chemical and a target at the probability of >0.9.

### 2.2. Preprocessing

We excluded non-plants from the raw dataset. For instance, minerals (non-living things) and animals (including corals) were removed, but the fungi were not. After excluding the non-plants, the raw data sets containing 5500 instances of plants and non-plants were reduced to a set containing 3614 plants. Then, the duplicated instances were removed based on their identical “scientific name,” “Latin name” and “use target”, which led to a remaining list of 2886 plants in the data set.

The multipartite network of the medicinal plants, chemicals, and targets was constructed by combining the two bipartite networks: the plants-chemicals network and chemicals-targets network (Figure 1A). The nodes in the network represented the plants, chemicals, and targets, and the edges in the network represented the relationships between the plants and chemicals, and between the chemicals and targets. From the multipartite network, we removed the nodes of degree zero, i.e., the plants, chemicals, or targets that were not linked to any other entities. After all, 1138 plants, 10,043 chemicals, and 441 targets remained for the combined multipartite network. The basic statistics of the preprocessed network data are summarized in Table 1.

### 2.3. Methods

We analyzed the multipartite network using the spectral co-clustering algorithm [35]. The algorithm clusters the medicinal plants, chemicals, and targets by solving a network partitioning problem. It partitions the set *V* of the connected nodes into *k* subsets, $V_{1},V_{2},\ldots ,V_{k}$ in such a way that the *k* subsets approximately minimize the normalized cut criterion:(1)$$Q(V_{1},V_{2},\ldots ,V_{3})=\frac{1}{k}\sum_{i=1}^{k}\frac{\mathrm{cut}(V_{i},\bar{V_{i}})}{\mathrm{vol}(V_{i})}$$
where $\bar{V_{i}}=V\backslash V_{i}$ is the complement of *V_i_*, the $cut\left(M,N\right)=\underset{m\in M,n\in N}{\sum }w_{mn}$ of two disjoint sets *M* and *N* is defined by the sum of the edge weights, $w_{mn}$, between two nodes from each of *M* and *N*, and $vol\left(V_{i}\right)=\underset{m\in V_{i}}{\sum }d_{m}=\underset{m\in V_{i}}{\sum }\underset{n}{\sum }w_{mn}$ the sum of degrees of all nodes in *V_i_*. The normalized cut criterion represents the level of connectivity between the *k* subsets, where a smaller value indicates a weaker connectivity between them.

To determine the number of clusters, *k*, we computed the modularity, defined by
(2)$$\mathrm{Modularity}(k)=\frac{1}{2E}\overset{k}{\underset{i=1}{\sum }}\underset{m\in V_{i},n\in V_{i}}{\sum }\left(w_{mn}-\frac{d_{m}d_{n}}{2E}\right)$$
where *E* is the number of edges and *d_m_* is the degree of a node *m* [37,38,39]. The modularity measures the strength of a partition with respect to the distribution of edges because it compares the number of edges, $\underset{m,n}{\sum }w_{mn}$ with the expected number of edges placed at random, $\underset{m,n}{\sum }d_{m}d_{n}/2E$, within the partition. An optimal number of clusters is determined at the maximum point of the modularity.

The hierarchical clustering analysis led to the identification of the clusters of medicinal plants based on their chemical profiles. The goal of the analysis was to provide insight into the biologically or chemically inter-related plants and search for their potentials medicinal effects in the integrated network. We use the Bray-Curtis distance for the hierarchical clustering, which is defined by, between two plants *u* and *v*,
(3)$$d\left(u,v\right)=\frac{\sum_{j}\left|u_{j}-v_{j}\right|}{\sum_{j}\left|u_{j}\right|+\left|v_{j}\right|}$$
where *u_j_* (or *v_j_*) is equal to one if the plant *u* (or *v*) is connected to the *j*th chemical in the network, and zero otherwise. The Bray-Curtis distance takes a value between 0 and 1, and it has two favorable properties for the multipartite network analysis: (i) It is a scale-free distance metric, that is, it depends only on the fraction of the number of chemicals connected to the plants, rather than the total number of chemicals in the network. The scale-freeness is a crucial property for analyzing sparsely connected networks, such as the multipartite network in this study where most of the *u_j_* (or *v_j_*) values are zero; see, for example, Appendix A for a visualization. (ii) It is suitable for the unweighted networks because it only considers the number of edges while ignoring the edge weights.

We used the complete linkage method for hierarchical clustering, that is, the distance between two clusters is defined by the distance between two elements that are farthest from each other. The combined use of the Bray-Curtis distance and complete linkage yields a meaningful interpretation of clustering results. That is, if we combine two clusters one whenever they are at a distance of less than one, then every pair of elements will have a distance less than one. It implies that every pair of medicinal plants in a cluster will have at least one common chemical because *d*(*u*,*v*) < 1 implies that *u_j_* = *v_j_* = 1 for some chemical *j*.

Finally, we proposed an index called TPS, which is a similarity measure between a plant cluster and a target. TPS of a plant cluster *P* to a target *T* is defined by the fraction
(4)$$\mathrm{TPS}(P,T)=\frac{\left|C_{P}\cap C_{T}\right|}{{|C}_{T}|}=\frac{\mathrm{number}\;\mathrm{of}\;\mathrm{chemicals}\;\mathrm{connected}\;\mathrm{to}\;\mathrm{both}\;P\;\mathrm{and}\;T\;}{\mathrm{number}\;\mathrm{of}\;\mathrm{chemicals}\;\mathrm{connected}\;\mathrm{to}\;T}$$
where *C_P_* is the set of all chemicals connected to at least one plant in the plant cluster *P*, *C_T_* is the set of all chemicals connected to the target *T*, and |∙| denotes the number of elements in a set. TPS can represent potential efficacy of a plant cluster for predicting a protein target. We used the TPS to search for the multipartite network for plant clusters of the highest efficacy or most active targets.

## 3. Results

### 3.1. Multipartite Network Analysis

We constructed the multipartite network with the nodes of medicinal plants, chemicals, and targets for the interpretation of the biological roles of these chemicals and plants containing them on multi-targets. The preprocessed network is shown in Figure 1B, where 10% of the nodes are randomly selected for visualization. Note that the plants and targets have connections only to the chemicals. Figure 1C shows the distribution of the node degrees for the plants, chemicals, or targets, where the vertical axes are on a log scale. The distribution of the power-law degree imply that the network is a scale-free network [40,41,42].

The structural properties of the multipartite network were further analyzed by the network analysis techniques. We divided the network into five subnetworks by applying the spectral co-clustering algorithm. Then, the structural properties of the multipartite network and the five subnetworks were computed and summarized in Appendix A. The computed structural properties are a standard set of theoretic graphical measures, such as the diameter, average bipartite clustering coefficient, and degree of assortativity. Appendix A illustrates the multipartite network, modularity, and five subnetworks, where only 10% of the whole nodes were randomly selected for visualization.

### 3.2. Hierarchical Clustering Analysis of Plants

We performed the hierarchical clustering analysis of the medicinal plants in the multipartite network, and generated dendrograms to visually inspect the clusters of the medicinal plants (Figure 2 and Appendix A). In the two largest clusters of the plants (Figure 2A,C, Appendix A), the same species with different usage parts have been aggregated at the bottom level (i.e., at the height of zero) in the dendrogram, and different species belonging to the same family are grouped together at low levels. However, some species belonging to the same family are far apart or the ones belonging to a different family are closely located. In the largest cluster, many different species sharing β-sitosterol, which is one of the most common phytosterols and is distributed over a wide range in the plant kingdom, existed [43]; many plants belonging to different families share more chemicals than the ones belonging to the same family (Figure 2A). The chemical diversity of the cluster shows that the specialized distribution of the secondary metabolites in plants is not dependent on their family level. However, in the second largest cluster, the chemical profiles of the plants depending on the family contributed to the classification of the smaller clusters (Figure 2C). Additionally, the chemical diversity of the plants in the second largest cluster (123 types of subclasses and 311 types of direct parent levels) was bigger than the largest cluster (109 types of subclasses and 257 types of direct parent levels) (Figure 2B,D). Among the smaller clusters, two clusters containing the specialized types of chemicals, protoberberine alkaloids (Figure 2E,F, and Appendix A) and vitamins (Figure 2G,H, Appendix A), were clearly categorized by the family. We expect that the clustering information by the chemical profiles between the plants can be utilized not only for the chemotaxonomic study of the medicinal plants, but also for alternative use of the medicinal plants.

### 3.3. Interaction between Plants and Targets Based on TPS

In the multipartite network, the interaction between the plants and targets is mediated by the connectivity of the chemicals in the network. We used the TPS to quantitatively measure the interaction that represents the potential efficacy of the plants for the targets. Because the plants in the same cluster share chemicals in common, the TPS was calculated between a cluster of plants and target.

The TPS was applied in two ways. First, the targets with the highest TPS values were selected for each of the four clusters of plants that were mentioned in the previous section and were manually investigated to validate our approaches. The subnetworks of the selected targets, chemicals connected to the targets, and corresponding cluster of plants connected to the chemicals are illustrated in Figure 3A–D, for the inspection of the effects of the mixture of chemicals or the plants in the cluster. Specifically speaking, the targets with the TPS values greater than 0.05~0.5 were selected while the number of chemicals connected to the targets was kept sufficiently large enough (at least 20) to produce reliable TPS values.

In each network, a single target had multiple edges with chemicals, but only a few chemicals contributed to the connection between the plants and targets. In the largest cluster, six targets, signal transducer and activator of transcription 3, phosphodiesterase 10A, anandamide amidohydrolase, prostanoid EP2 receptor, sphingosine 1-phosphate receptor Edg-3, and pepsin A, are displayed (Figure 3A). For example, in case of phosphodiesterase 10A, which has been studied as a therapeutic target for colon cancer and neurological symptoms [44], 12 chemicals, mainly carboline-type alkaloids, are displayed with linkages to the target (Appendix A). The well-known targets, such as cannabinoid CB1 and CB2 receptors, in the network from the second largest cluster (Figure 2C and Figure 3B) shared the monoterpenoids and benzopyran-type chemicals including cannabinoids from the genus Cannabis [45]. In the third network, protoberberine-type alkaloids from three families, Papaveraceae, Ranunculaceae, and Rutaceae, were linked with the targets, dopamine receptors, and alpha adrenergic receptors, which are known as the main target of these chemicals [46,47]. Tetraterpenoids, such as lycopene, lutein, and β-carotene, which are the well-known chemicals targeting the retinoid signaling pathway, had multiple linkages with retinoic acid receptor gamma (RAR-γ) and retinoid X receptor beta (RXR-β) [48,49,50]. These results show that the target information predicted by the chemicals in the multipartite network is reliable enough to investigate alternative targets of the chemicals, for which biological roles have not been defined in previous studies or they existed as mixed forms with other chemicals in a plant.

Second, the proposed TPS was applied for searching the closest related plants, which are predictive of a given target based on the highest TPS value. The results provide clues to understand the complex biological roles of the combination of medicinal plants or the chemicals produced from the plants. In this scenario, a cluster of plants is recommended from among the large-scale network as the candidate medicinal plant for a single target of interest. For example, we used ”tyrosinase” as the model target to validate the suggested method and to investigate the medicinal effects of the retrieved cluster of plants (Figure 4A). In the multipartite network retrieved for the target “tyrosinase,” 165 chemicals from 31 different species were linked with the target node. Among them, 50 chemicals from Morus alba were found in the middle of the network (Appendix A). Additionally, *Carthamus tinctorius*, *Cudrania tricuspidata*, *Glycyrrhiza inflata*, and *Glycyrrhiza uralensis* had more than 10 chemicals, which were predicted to act on tyrosinases (Appendix A). These five plants have been known as the most popular and successful sources for the whitening products used in the cosmetic industry [51,52].

Next, we combined three subnetworks-networks from three targets, steroid 5α-reductase 1 (S5R1), steroid 5α-reductase 2 (S5R2), and androgen receptor (AR), which are involved in the androgen metabolism, for the interaction of plants and chemicals on multi-targets (Figure 4B). In total, 44 among 52 and 54 chemicals from S5R1 and S5R2, respectively, shared with each other, but 10 of 52 from S5R1 and 7 of 54 from S5R2 only shared those with 58 chemicals from AR (Appendix A). The types of chemicals between the pairs of targets showed significantly different patterns. The chemicals co-targeting S5R1 and S5R2 were mainly the steroid derivatives, such as cholestanes, ergostanes, and stigmastanes, which were synthesized via the triterpenoid pathways (Figure 4); however, sesquiterpenoids and diterpenoids were the main components sharing the pairs of S5R1 and AR, or S5R2 and AR (Figure 4). We found that only 6 chemicals simultaneously acted on three targets and plants containing them (Table 2).

## 4. Discussion

In this study, we tried to unravel the complex interaction between multi-chemicals and multiple targets using the computational target prediction model and multipartite networks comprising the relationship between the plants, chemicals, and targets. In hierarchical cluster analysis, plants producing similar specialized metabolites are clustered only within a taxon. In particular, in chemotaxonomy, secondary metabolites can be used as specialized markers to distinguish among plants [53]. However, not all secondary metabolites are produced in the same taxon. For example, flavonoids are widely distributed in plants and even in fungi, fulfilling many functions, such as ultraviolet filtration, symbiotic nitrogen fixation, and floral pigmentation [54]. The chemical profile of each plant has a minor effect on the classification of the plant.

To unravel the clusters of multipartite networks from hierarchical clustering analysis, we introduced an index of potential efficacy of plant clusters toward a target, called TPS. TPS indicated the proximity of subnetworks of the metabolic relationships between plant clusters and targets. We utilized it to search for and suggest candidate plants and chemicals acting on specific targets (Figure 3). It was validated that the plants with more than 10 chemicals had enough experimental evidence to explain the biological roles on specific targets retrieved in the network. TPS was also used for the extraction of a list of plants and chemicals acting on a specific target of interest (Figure 4).

We also obtained potent plant and chemical candidates for the specific target, “tyrosinase”, confirming the validity of the approach. The node information of the plants in the network is expected to guide toward new alternative sources developed by plant combinations.

The approach is based on the multipartite network of plants and chemicals for a single target using the TPS and could be expanded for multi-targets that closely interact with each other. The plants and chemicals, which are expected to act on multiple targets, including S5R1, S5R2, and AR, could be suggested to control the levels of testosterone (T) and dihydrotestosterone (DHT), involved in the androgen metabolism. It is known that the blockade of AR can decrease benign prostate hyperplasia (BPH) symptoms and further relieve the lower urinary tract symptoms [55]. Since the alterations in AR are associated with prostate cancer, the inhibition of AR by drugs is the prevention target for prostate cancer growth [56]. The 5α-reductase inhibitors, which suppress the conversion of T into DHT, contribute to the reduction in BPH or preventing prostate cancer [55,57]. In the present study, the AR-targeted chemicals were sesquiterpenoid and diterpenoid-type, while the steroid derivatives were suggested as the chemicals targeting 5α-reductases (S5R1 and S5R2). We expected that the plants containing the chemicals acting on all three targets, S5R1, S5R2, and AR, can be the new candidates for controlling the symptoms or diseases involving androgen metabolism, such as BPH and prostate cancer. The well-organized multipartite network is expected to explain the synergistic effects between the multi-chemicals in single or multiple medicinal plants.

We have constructed a multipartite network by integrating the databases of the medicinal plants, chemicals, and targets for the prediction of the biological roles of the plants and chemicals on specific targets. We have presented a novel computational approach based on the multipartite network with which the plant species can be distinguished based on their chemical profiles. In the clustered bipartite network of plants and chemicals, the chemical profiles of the plants in identical or similar species were clustered closely at the bottom level of the dendrogram. However, the plants in the same family were found far apart in the dendrogram in many cases, which implies that the chemical profiles of the plants do not significantly contribute to the classification of the species.

## 5. Conclusions

We have demonstrated that the network analysis helps to understand the biological roles of multi-chemicals on multi-targets. With the target prediction models, it also provides a way to discover alternative therapeutic agents from natural sources by extrapolating from the known bioactivities of chemicals on targets. The predicted results about the biological roles of multi-chemicals on multi-targets allow to explain the therapeutic activity of the medicinal plant extracts via subnetworks of molecular mechanisms and suggest the combinatory preparation of several medicinal plants on multi-targets to be involved in the symptoms or diseases of interest. In our future study, we will study the candidates, especially the combinations of plants or chemicals, predicted from the multipartite network in in vitro and in vivo models to validate our approaches and further develop novel therapeutic botanical drugs.

## Figures and Tables

**Figure 1 biomolecules-11-00546-f001:**
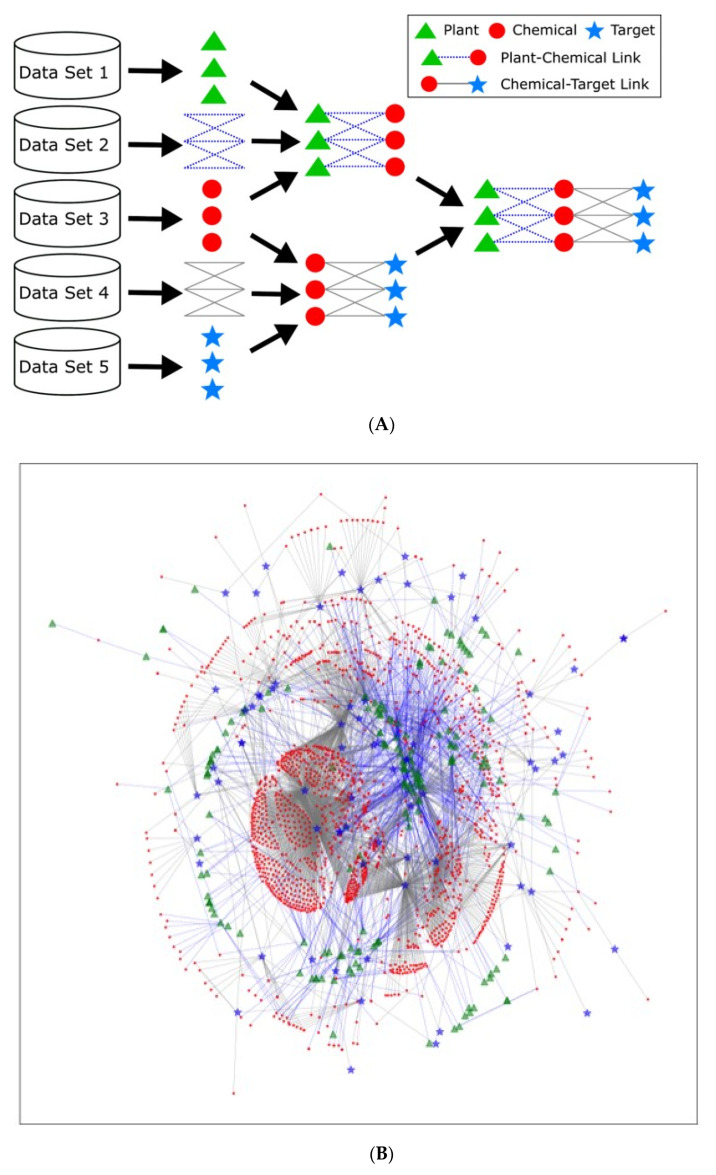
(**A**) The construction and visualization of the multipartite network database. (**B**) The representative multipartite network generated from the nodes which were randomly selected from 10% of the nodes 1138 plants, 10,043 chemicals, and 441 targets in the whole network. (**C**) The distribution of the node degrees for the plants, chemicals, or targets.

**Figure 2 biomolecules-11-00546-f002:**
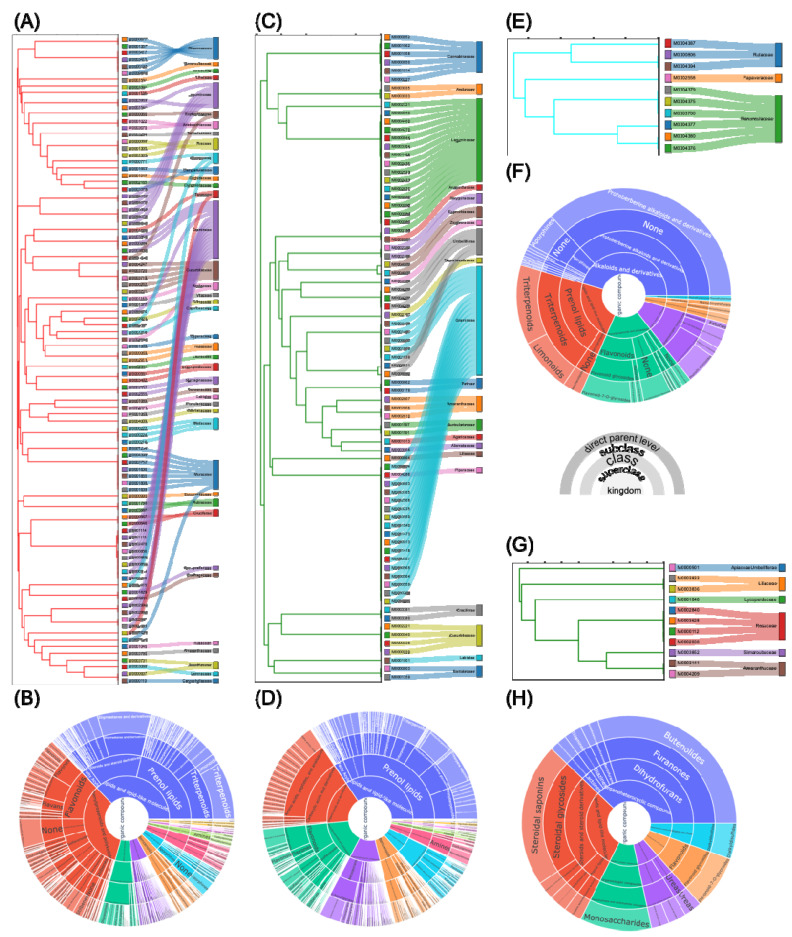
The four sub-clusters in the whole dendrogram (Appendix A). (**A**) The largest sub-cluster in the whole network. (**B**) The classification of chemicals in (**A**). (**C**) The second largest sub-cluster in the whole network. (**D**) The classification of chemicals in (**C**). (**E**) The sub-cluster of the nodes sharing protoberberine-type chemicals in the whole network. (**F**) The classification of chemicals in (**E**). (**G**) The sub-cluster of the nodes sharing vitamins in the whole network. (**H**) The classification of chemicals in (**G**).

**Figure 3 biomolecules-11-00546-f003:**
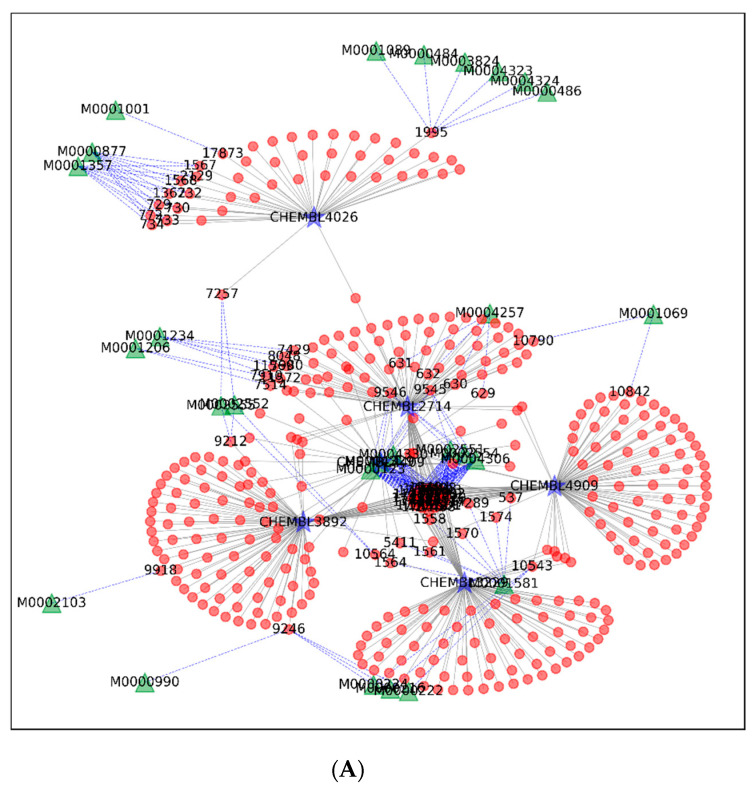
(**A**) The network diagram for the plants cluster in Figure 2A and the 6 targets retrieved based on the TPS index greater than or equal to 0.1. The chemicals effective to the targets are depicted in red circles. (**B**) The network diagram for the plants cluster in Figure 2C and the 2 targets retrieved based on the TPS index greater than or equal to 0.5. The chemicals effective to the targets are depicted in red circles. (**C**) The network diagram for the plants cluster in Figure 2E and the 7 targets retrieved based on the TPS index greater than or equal to 0.05. The chemicals effective to the targets are depicted in red circles. (**D**) The network diagram for the plants cluster in Figure 2G and the 5 targets retrieved based on the TPS index greater than or equal to 0.05. The chemicals effective to the targets are depicted in red circles.

**Figure 4 biomolecules-11-00546-f004:**
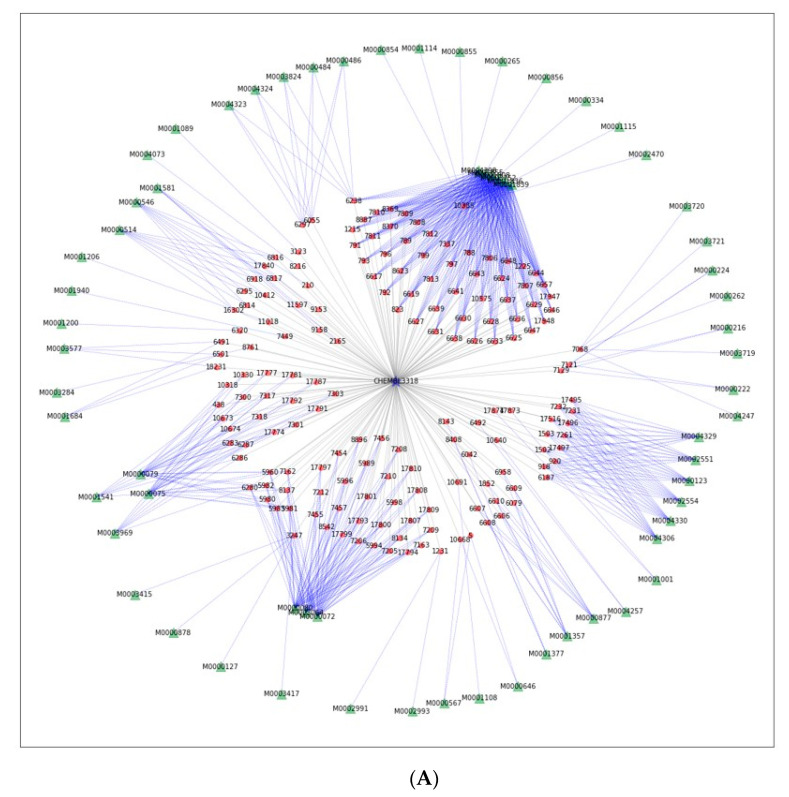
(**A**) The network diagram for the plants cluster having the largest TPS index to the target ”Tyrosinase” (CHEMBL3318). (**B**) The combined network generated from three network diagrams for the plants cluster having the largest TPS index to the three targets, ”steroid 5α-reductase 1“ (S5R1, ChEMBL1787), ”steroid 5α-reductase 2“ (S5R2, ChEMBL1856) and ”androgen receptor” (AR, ChEMBL1871). The chemical profiles sharing from chemicals from S5R1 and S5R2, from S5R1 and AR and from S5R2 and AR.

**Table 1 biomolecules-11-00546-t001:** Basic statistic of the combined multipartite network data.

	Number of Nodes				Number of Edges			Density ^1^
	Plant (Np)	Chemical (Nc)	Target (Nt)	Total (N)	Plant-Chemical	Chemical-Target	Total (E)	
Raw Data	5500	10,056	1224	16,780	58,068	100,290	158,358	0.00234
without Non-plants	3614	10,056	1224	14,894	54,585	100,290	154,875	0.00318
without Non-plants or Duplicates	2886	10,056	1224	14,166	34,549	100,290	134,839	0.00326
Only targets with the probability of above 0.9	2886	10,056	441	13,383	34,549	73,112	107,661	0.00322
Pre-processed Data	1138	10,043	441	11,622	34,549	73,112	107,661	0.00679

^1^ The density of a multipartite network is computed by E/(Np × Nc + Nc × Nt).

**Table 2 biomolecules-11-00546-t002:** List of chemicals simultaneously acting on three targets, S5R1, S5R2 and AR.

Chemical Name	CAS Number	Chemicals Classification				
		Kingdom	Superclass	Class	Subclass	Direct Parent
Kusunol	20489-45-6	Organic compounds	Lipids and lipid-like molecules	Prenol lipids	Sesquiterpenoids	Eremophilane, 8,9-secoeremophilane and furoeremophilane sesquiterpenoids
Stigmasta-4,6-dien-3-one	29374-98-9	Organic compounds	Lipids and lipid-like molecules	Steroids and steroid derivatives	Stigmastanes and derivatives	Stigmastanes and derivatives
Stigmastan-3-one (5-alpha)	102734-69-0	Organic compounds	Lipids and lipid-like molecules	Steroids and steroid derivatives	Stigmastanes and derivatives	Stigmastanes and derivatives
Dehydroabietinal	13601-88-2	Organic compounds	Lipids and lipid-like molecules	Prenol lipids	Diterpenoids	Diterpenoids
(-)-alpha-Copaene	3856-25-5	Organic compounds	Lipids and lipid-like molecules	Prenol lipids	Sesquiterpenoids	Sesquiterpenoids
(-)-Ylangene	14912-44-8	Organic compounds	Lipids and lipid-like molecules	Prenol lipids	Sesquiterpenoids	Sesquiterpenoids

## Data Availability

The data presented in this study are available on request from the corresponding author.

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
