# Peer review of "Cluster Analysis of Medicinal Plants and Targets Based on Multipartite Network"

_biomolecules, 2021, doi:10.3390/biom11040546_

Round 1

Reviewer 1 Report

As an old organic chemist, I’m still interested in drug discovery. That is why I accepted to review the publication of H. Yang et al. I add also to confess that I’m ignorant about hierarchical clustering and multipartite network.

Thus, I will only make comments on chemistry and I would suggest the editor add another reviewer.

  • In the introduction, a citation of the recent review on “Ethnobotany and the role of plant natural products in antibiotic drug discovery” by Cassandra L. Quave, Chem. Rev. 2021, 121, 3495−3560 could be fine.
  • In table 2, the authors use their approach to discovered alternative molecules to interact with 3 targets, i.e. steroid 5-alpha-reductase 1, steroid 5-alpha-reductase 2, and androgen receptor. However, I was a little frustrated by the lack of variability of the structures. The six molecules listed in table 2 are terpene-based molecules, with high log P. Why such low diversity from a database containing up to 1’879’206 compounds?
  • In conclusion, the authors mentioned that in their future study, these molecules will be studied. Did these molecules (listed in table 2) have already been studied in the literature?
  • The use of the ChEMBL number is confusing for the chemical reader. I know that I could find what is related to a ChEMBL number and I could understand why it is easy for the authors to use such a number. But, for me, “steroid 5-alpha-reductase 1” or even “SR type 1” is more meaningful than a ChEMBL number. As such, I suggest in the text to remove the ChEMBL number. These numbers could be found and use in the Supp Info.
  • The use of the nomenclature “inchikey” is confusing for the reader (see table 2).

As an example, I try to determine the structure of the product Chemical ID = 9654 (page 15), named by the authors “(-)-Ylangene”.

By using InChI resolvers like the UniChem service, I could transform

VLXDPFLIRFYIME-UHFFFAOYSA-N

into Pubmed PubChem CID N°19725. Then I found that the name is (-)-a-Copaene and I could found this molecule has two CAS numbers, i.e. 3856-25-5 and 138874-68-7! I also found that a synonym is a-ylangene not “(-)-Ylangene” as mentioned by the authors.

From the Scifinder website, I found that the correct CAS N° is 3856-25-5 with absolute stereochemistry defined. The other name for this molecule are: (-)-Copaene; (-)-α-Copaene; Aglaiene; alpha-Copaene and α-Copaene and a-ylangene was not found.

Thus, I think that the correct name and CAS N° instead of inchikey should be provided by the authors.

Author Response

We appreciate the careful and thoughtful review given to our manuscript. We try to prepare the suitable responses for your considerable comment as below and also reflect your comments in our manuscript 

  • In the introduction, a citation of the recent review on “Ethnobotany and the role of plant natural products in antibiotic drug discovery” by Cassandra L. Quave, Chem. Rev. 2021, 121, 3495−3560 could be fine.--> Really appreciate it. We added this reference in the introduction section.
  • In table 2, the authors use their approach to discovered alternative molecules to interact with 3 targets, i.e. steroid 5-alpha-reductase 1, steroid 5-alpha-reductase 2, and androgen receptor. However, I was a little frustrated by the lack of variability of the structures. The six molecules listed in table 2 are terpene-based molecules, with high log P. Why such low diversity from a database containing up to 1’879’206 compounds?

        -> In the study, we used 10,056 chemicals in KTKP data sets, not 1,879,206 compounds in ChEMBL database. 10,056 chemicals were isolated in 5,500 plants in KTKP data sets and have the edges with their plants. Thus, terpene-based molecules were suggested in the results.

  • In conclusion, the authors mentioned that in their future study, these molecules will be studied. Did these molecules (listed in table 2) have already been studied in the literature?

        -> We already tried to search those compounds in literature, but could not unfortunately find them. They are rarely found in nature, but worth to be studied with the expectation of the bioactivity.

  • The use of the ChEMBL number is confusing for the chemical reader. I know that I could find what is related to a ChEMBL number and I could understand why it is easy for the authors to use such a number. But, for me, “steroid 5-alpha-reductase 1” or even “SR type 1” is more meaningful than a ChEMBL number. As such, I suggest in the text to remove the ChEMBL number. These numbers could be found and use in the Supp Info.

        -> We agree the reviewer’s comment and deleted them in the main text. Since they were annotated in the figures, they remained in the legends of figures.

  • The use of the nomenclature “inchikey” is confusing for the reader (see table 2). As an example, I try to determine the structure of the product Chemical ID = 9654 (page 15), named by the authors “(-)-Ylangene”. By using InChI resolvers like the UniChem service, I could transform VLXDPFLIRFYIME-UHFFFAOYSA-N into Pubmed PubChem CID N°19725. Then I found that the name is (-)-a-Copaene and I could found this molecule has two CAS numbers, i.e. 3856-25-5 and 138874-68-7! I also found that a synonym is a-ylangene not “(-)-Ylangene” as mentioned by the authors. From the Scifinder website, I found that the correct CAS N° is 3856-25-5 with absolute stereochemistry defined. The other name for this molecule are: (-)-Copaene; (-)-α-Copaene; Aglaiene; alpha-Copaene and α-Copaene and a-ylangene was not found. Thus, I think that the correct name and CAS N° instead of inchikey should be provided by the authors.

        -> We agree the reviewer’s comment. We revised all of the names in tables and added CAS numbers instead of InChI codes. Actually, the information for chemicals using data sets in this study consists of names,  such as Ylangene and Copaene. We really appreciated your suggestions, and will use the CAS number instead of InchIkey

Reviewer 2 Report

In presented publication valuable analysis of biologically and chemically related actions of plant active bodies was conducted. Searching for potential medicinal effects was done. Analysis of integrated network and hierarchical clustering analysis were performed.

Investigations of complex interaction between multi-chemicals and multiple targets using the computational target prediction model and multipartite networks comprising the relationship between the plants, chemicals and targets are planned to be used in future study to create novel therapeutic botanical drugs.

Authors may consider suggestions given below to improve their publication:

  1. Conclusions could be more detailed - such investigations are planned by authors. New multicomponent plant drugs can be proposed to act on one physiological target. If the authors could give some example it would improve the value of presented work.
  2. In the first line of Table 1 headlines may be put in one line.

Author Response

We appreciate the careful and thoughtful review given to our manuscript. We try to prepare the suitable responses for your considerable comment as below and also reflect your comments in our manuscript

1. Conclusions could be more detailed - such investigations are planned by authors. New multicomponent plant drugs can be proposed to act on one physiological target. If the authors could give some example it would improve the value of presented work.

→ We are studying the combination of plants predicting on the specific targets using our approach. Thus, we had to mention it simply in this study.

2. In the first line of Table 1 headlines may be put in one line.

→ We revised the title of Table 1 as ‘Basic statistic of the combined multipartite network data.“